# On the dimensionality of odor space

Markus Meister*

**Division of Biology and Biological Engineering, California Institute of Technology, Pasadena, United States**

**Abstract** There is great interest in understanding human olfactory experience from a principled and quantitative standpoint. The comparison is often made to color vision, where a solid framework with a three-dimensional perceptual space enabled a rigorous search for the underlying neural pathways, and the technological development of lifelike color display devices. A recent, highly publicized report claims that humans can discriminate at least 1 trillion odors, which exceeds by many orders of magnitude the known capabilities of color discrimination. This claim is wrong. I show that the failure lies in the mathematical method used to infer the size of odor space from a limited experimental sample. Further analysis focuses on establishing how many dimensions the perceptual odor space has. I explore the dimensionality of physical, neural, and perceptual spaces, drawing on results from bacteria to humans, and propose some experimental approaches to better estimate the number of discriminable odors.

## Introduction

The perceptual space for human color vision has three dimensions. Experimental proof of this dates to the 17th century, when it was found that every color sensation can be matched by mixing together three primary lights, but not if only two lights are available (**Mollon, 2003**). Therefore every color sensation can be fully characterized by three numbers, namely the intensity of the primaries that match it. We now know that color vision is based on three kinds of cone photoreceptors in the retina that differ in their sensitivity to the wavelength spectrum of light. Any pattern of excitation among cones can be matched by an appropriate mix of three primary lights, and this is the basis for RGB color display devices. Cone signals get processed by several circuits in the retina and beyond. This system ultimately has limited resolution, and in practice human subjects can distinguish about 1–2 million different colors (**Masaoka et al., 2013**). Clearly, a quantitative understanding of color perception has both energized the search for the underlying neural circuits and made possible the design of image display technologies that mimic reality.

One would like to achieve a similarly satisfying understanding of human smell. Human olfaction begins with the binding of odor molecules to olfactory receptors, of which there exist ~400 types (**Malnic et al., 2004**). It is believed that these receptor types all differ in their relative sensitivity to various odorants. Two odors can only be distinguished if they cause different patterns of activity among these types. Thus the neural space of odors at the very input to the olfactory system would seem to have 400 dimensions, many more than encountered in the color system. But what is the perceptual space for odors? The exact analog of the early color mixing experiments has not been done, but from a quantitative analysis of perceptual similarities it has been argued that the space of odors is dominated by just one or two dimensions (**Secundo et al., 2014**), much fewer than the 400 dimensions at the level of sensory receptors. Similarly the number of discriminable odors is thought to be only 10,000, although this estimate is largely anecdotal (**Gilbert, 2008**). Clearly there is a paradox that remains to be resolved pitting the perceptual space of human smell against the receptor space.

On this background, a recent article claims that 'Humans can discriminate one trillion odors' (**Bushdid et al., 2014**). If correct, this would dramatically reorient thinking in this field. Not least, such

*For correspondence: meister@caltech.edu

**Competing interests:** The author declares that no competing interests exist.

**eLife digest** Scientists are interested in the number of colors, sounds and smells we can distinguish because this information can shed light onto how our brains process these senses both in health and disease. It is relatively straightforward to determine how many colors we can see or sounds we can hear because these stimuli are well defined by physical properties such as wavelength. We know the range of wavelengths that the eye can see or the ear can hear, and we can also understand how two such stimuli (e.g., red and blue) are arranged perceptually (think of a color wheel). It is harder, however, to do the same for smell because most 'olfactory stimuli' consist of mixtures of different odor molecules. Moreover, we understand much less about how olfactory stimuli are arranged perceptually.

In 2014 researchers at Rockefeller University reported that humans can distinguish more than one trillion smells from one another. To calculate this number the researchers tested the ability of human subjects to discriminate between mixtures of different odor molecules. Each mixture consisted of 10, 20 or 30 molecules selected from a chemical library of 128 different odor molecules. Since each mixture of 10 molecules could contain any 10 of the 128 molecules, more than 200 trillion combinations were possible; the number of possible combinations for the 20- and 30-molecule mixtures were even higher.

The aim of the experiment was to identify—by sampling from this very large number of combinations—the number of molecules that two mixtures could have in common and still be distinguishable to the typical person. The Rockefeller team used this number and a geometrical analogy to conclude that humans could discriminate at least 1.72 trillion odors, which was much higher than expected from previous reports and anecdotes.

Now Meister reports that the claims made in the Rockefeller study are unsupported because of flaws in the design and analysis of the experiment. In particular, there are flaws in the mathematical methods used to infer the potential number of all smells that humans can discriminate from the numbers of experimental samples tested. Meister also applies the Rockefeller approach to a well-understood sensory system—the vision system—and finds that it predicts that humans should be able to discriminate an infinite number of colors: however, it is widely agreed that humans can only discriminate several million colors.

In a separate paper Gerkin and Castro also report that the 1.72 trillion smells claim is unjustified.

a result would dash any realistic hopes for olfactory displays that can mix any odor sensation from a small number of primaries. The result was nominated for 'Breakthrough of the Year 2014' and heavily promoted to the popular press. By now many people 'know' that humans discriminate a trillion odors, and that our color vision system pales miserably in comparison.

Remarkably, these claims were based on an experimental study in which humans discriminated successfully only 148 pairs of odors. Clearly some mathematical method was needed to extrapolate from that number to trillions. Here I show that this mathematical method fails, and as a result the claims are without any basis. We will see that the data on human olfaction are equally consistent with a trillion discriminable odors and with just 10 (and anything in between). Moreover, if the same method were applied to human color vision, one would conclude that humans can distinguish at least $10^{27}$ colors, in dramatic conflict with experimental evidence. Beyond correcting the erroneous claims, the analysis of failure in the Bushdid et al. study yields useful insights about the nature of the olfactory code. I follow with some suggestions for an approach to estimate the number of distinguishable odors.

## Results

### How not to estimate the number of discriminable stimuli

The reader is encouraged to read the original report by *Bushdid et al. (2014)*. The following is a brief summary of the procedures used in that study. The space of all possible odor stimuli is huge. There are likely several hundred thousand distinct chemicals that smell. Any mixture of those chemicals is a point

in odor space. The goal is to find how many of those mixtures produce distinct sensations. What is the largest collection of odors such that any one is discriminable from all the others?

The reported experiments were limited to a subspace spanned by 128 substances. From these primary odors the authors made mixtures by drawing a number ($N$) of primary odors and mixing those in equal parts. Each primary odor is either present or absent in the mixture. Some of these mixtures are very similar to each other, for example if they share 29 of the 30 primary components. Others are very different, for example when they share none of the primary components. For any two mixtures the number of unshared components can be defined as their 'distance'. The authors assume that the ability of humans to discriminate two odors improves systematically with this distance.

The next step is to determine the critical distance at which two odors become discriminable. The authors probe this systematically by making pairs of mixtures with $N$ components of which $M$ are unshared, and testing those for discrimination by human subjects. Indeed they find that the probability of discrimination increases with $M$ (Figure 3B of *Bushdid et al., 2014*). They define the critical distance $D$ as that separation $M$ at which 50% of the mixture pairs are discriminable.

With these assumptions, two odors closer than $D$ tend to smell the same, whereas two odors separated by more than $D$ will smell different. To determine how many discriminable odors there are, the authors ask how many regions of diameter $D$ can fit in the original 128-dimensional odor space. This is analogous to the problem of packing spheres in high-dimensional spaces and the authors compute the number of packable spheres of diameter $D$ by methods of combinatorics (Figure 3D of *Bushdid et al., 2014*). This yielded the 'one trillion' in the title of their paper.

## Failure on a simple model with three odor responses

Given a novel analysis method, it can be instructive to test it on a 'simplest possible' model, for which the desired answer is known and the calculations are easy. In the present context we need a model of olfactory processing that treats mixtures of many components, can compare pairs of such mixtures, and exhibits a clear performance limit in their discrimination. Imagine a toy microbe that lives in an environment with many odors and has receptors to sense all of them. Pure odors are either attractants (providing a sensory input of +1) or repellents (−1). Following the procedure of *Bushdid et al. (2014)*, we choose at random 128 of those as primary odors and combine them to make mixtures containing 30 odors. The microbe responds to a mixture by simply summing the sensory input from the component odors. If the sum is less than −2 it says 'yuck'; if it is greater than +2 it says 'yum'; and from −2 to +2 it says 'meh'. Two odor mixtures are discriminable if the microbe responds differently to them.

Now we make odor mixtures that share a certain number of odors and plot the fraction of discriminable mixtures vs the odor overlap (*Figure 1A*, compare to Figure 3B of *Bushdid et al., 2014*). We find that the critical value of 50% discriminability is reached when the mixtures share 15 of the odors. This leads to an estimate of ∼9·10$^{11}$ discriminable odors (*Figure 1B,C*, see Figure 3D of *Bushdid et al., 2014*). So by the standards of the proposed analysis, this toy microbe can also discriminate 1 trillion odors. Yet we know by construction that it recognizes only three classes of odors. Thus one can produce at most three odors such that each can be discriminated from all the others.

One gets the sense that something is amiss with the analysis procedure of Bushdid et al. If it fails so dramatically on a crude toy model of odor integration, should one trust it on more complex sensory systems? Nevertheless it is instructive to evaluate the procedure on a more realistic case. For example, the toy microbe actually has a name for each odor in the space (one of three possible names) whereas the human subjects were not asked to name the odor. Furthermore, for any given mixture pair the microbe's response is deterministic, whereas humans varied in their response. Thus I will now test the analysis on a system in which one can more exactly replicate the human psychophysics methods used in the reported odor discrimination tests.

## Failure on the color vision system

A prominent claim in *Bushdid et al. (2014)* is that olfaction vastly outperforms color vision in terms of discriminable stimuli. Specifically the authors compare their own estimate of a trillion discriminable odors to the literature's estimates of ∼1 million discriminable colors. However, studies of color vision used a very different procedure to determine the number of distinct sensations (more on this below). Here I ask what number the authors' methods would produce. Fortunately we know enough about the

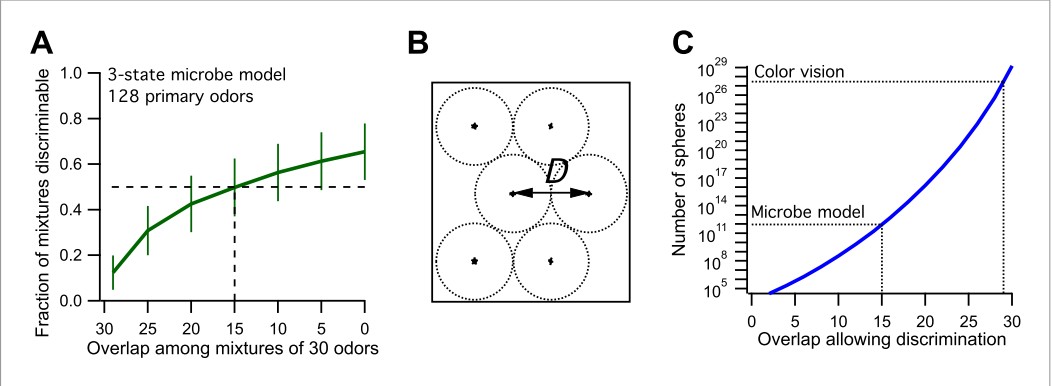

**Figure 1**. Model of olfaction in a toy microbe. (**A**) This 3-state olfactory system counts how many odors in the mixture are attractants vs repellents, and converts the result into three response categories (see text). Two odor mixtures are discriminable if they cause different responses. A numerical simulation of the response to many odor mixtures yields the fraction of discriminable mixtures as a function of the number of odors, $O$, that they share. Mean ± SD over 1000 repeats using different random assignments of the primary odors. Horizontal dashes: criterion for critical distance (50% discriminable pairs). Vertical dashes: critical distance $D = 30 - O = 15$. (**B**) Points in odor space separated by a distance $D$ cause different responses at least half the time. Counting how many such points exist in the space is like trying to pack spheres of diameter $D$ to fill the space as efficiently as possible. (**C**) The number of such spheres in 128-dimensional space as a function of the discriminable overlap $O$ among 30-odor mixtures, computed by the formula given in *Bushdid et al. (2014)*. The value $O = 15$ from panel (**A**) yields ∼9 × 10$^{11}$ spheres.

rules and mechanisms of color perception to simulate this with great confidence, so there is no need to actually perform new color discrimination trials.

The space of all colored lights has infinite dimensions. Each light is characterized by its wavelength spectrum $S(\lambda)$, which specifies how much power exists at each wavelength $\lambda$. Because the wavelength can take on any positive real value, the $S(\lambda)$ are functions of a continuous variable, and thus have infinite dimensions. However, as introduced above, the sensory space of human color vision has only 3 dimensions (*Wandell, 1995*). For each of the three cone types in the retina, the excitation is determined by the rate of photon absorption by its visual pigment. In turn, that rate is a linear function of the spectrum of the light, namely the projection of the light spectrum onto the absorption spectrum of the pigment. If two lights are mixed together, the resulting excitation is the sum of the effects of the individual lights. Therefore human color vision begins by projecting the infinite space of lights down onto a subspace of just 3 dimensions, spanned by the three cone excitations. This subspace has been probed extensively in psychophysical experiments. Typically the subject is shown two lights side-by-side and asked whether they appear different. By systematic probing of the 3-dimensional space one finds there are upward of 1 million discriminable lights, in the sense that any two of them will look different in a pairwise comparison (*Masaoka et al., 2013*).

For the purpose of simulation, I will therefore consider a space with three axes: $R$, $G$, and $B$ (*Figure 2A*), corresponding to the three cone excitations. For simplicity, the three variables will range from 0 (dark) to 1 (bright), so the space is a unit cube. Every physical light stimulus is projected onto a vector in this unit cube. The color vision system adds some noise to that vector along each of the 3 dimensions. As a result, two lights are discriminable if their vectors are separated by more than the noise. The level of noise is chosen so that a difference of 0.01 along any dimension is discriminable, which gives 1 million discriminable vectors in the unit cube.

Now one can implement the procedures of *Bushdid et al. (2014)*: Choose at random 128 primary lights in this space and make them of equal intensity. Then produce mixtures of 30 lights from those primaries; design different classes of mixtures that vary in the number of shared components. Within each mixture class, present pairs of stimuli to the model and ask whether they can be discriminated, following an 'odd-man-out' procedure. Plot the fraction of discriminable pairs against the mixture overlap. Find the overlap at which that fraction is 50%, and take the number of unshared components as the critical distance $D$ for discrimination. Compute how many spheres of size $D$ fit in the original 128-dimensional space.

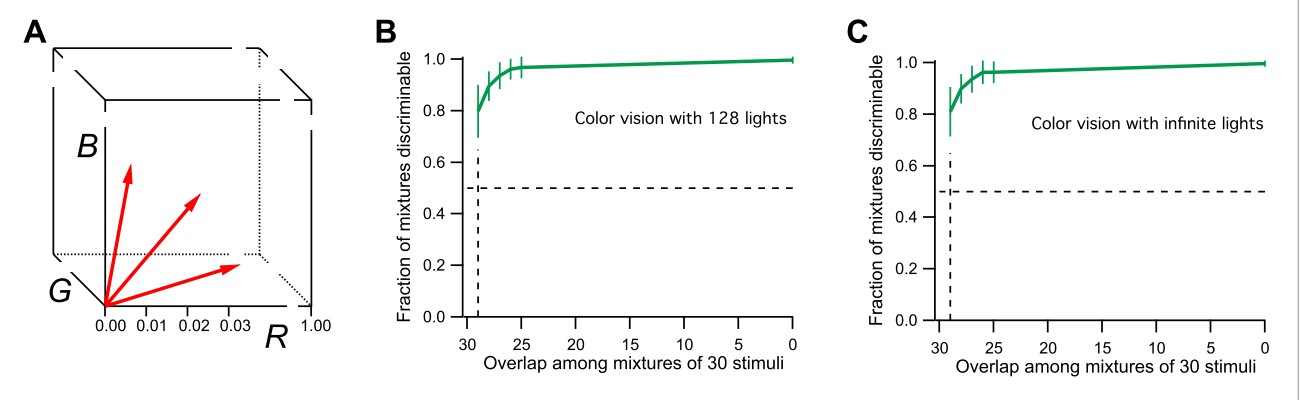

**Figure 2**. Model of human color vision. (**A**) The RGB color cube with three of the 128 primary colors represented by vectors from the origin. Tick marks represent just noticeable differences, for example, along the $R$-axis. (**B**) The fraction of discriminable 30-light mixtures as a function of their overlap. Mean ± SD over 1000 repeats using different random assignments of the 128 lights. 30 lights per mixture, 20 mixture pairs per class, 26 subjects per pair. Horizontal dashes: criterion for critical distance. Vertical dashes: critical distance. (**C**) As in panel **B** but with the mixture components drawn at random from all possible directions in the space rather than from a preselected set of 128 primaries. The results are almost identical.

From this simulation, I find that all 30-light mixtures are discriminable if they differ by as little as one component (*Figure 2B*). This is to be expected. Each of the primary lights has a vector length of 1/30 (see 'Materials and methods'). This means that a single unshared component can separate the two mixture vectors by more than 1/30 in the RGB space. But a separation of just 1/100 along any axis is discriminable. So the critical distance $D = 1$. From this one calculates there are more than $10^{27}$ discriminable colors in the 128-dimensional space probed here (*Figure 1C*, see Figures 3D and S1B of *Bushdid et al., 2014*).

Actually, the number of discriminable colors by this argument is much higher, even infinite. To drop down to the 50% discriminability criterion used by the authors in defining $D$, we have to make much larger mixtures of ~60 lights that differ by just one component. The number of possible mixtures of that kind is about $10^{37}$. Furthermore, there is no reason to limit the starting space to just 128 primary lights. There is an infinity of spectra that are physically achievable lights, so we could have started with a subspace of arbitrarily high dimensionality. One can simulate infinite dimensionality by choosing for each of the mixtures a different random set of 30 normalized vectors from the cube (again maintaining the specified overlap among mixture pairs). As shown in *Figure 2C*, the critical distance under those conditions is still 1. Thus the methods of *Bushdid et al. (2014)* would conclude that humans can discriminate an infinite number of colors. That is much bigger than 1 trillion, so color vision would still win over smell, at least by the internal logic of this analysis. However, the number is also much bigger than the known experimental result, a few million. We can conclude that the analysis method fails the 'positive control test', namely the application to a related problem with known solution.

## Humans can discriminate at least 10 odors

Given that the analysis of Bushdid et al. can vastly overestimate the number of discriminable stimuli, one wonders whether the human odor discrimination data in that report could equally be explained by a much smaller number of odor percepts. That is indeed the case.

Let us consider a simple model of odor processing in which there is a very large number of odor stimuli and a large number of associated odor receptors. However, I will suppose that the nervous system ultimately projects all those odors onto a neural representation with a single dimension, and we can use the vectors on the unit circle for that (*Figure 3A*). So the 128 primary odors map onto 128 unit vectors with random angles. Furthermore, we will suppose that a mixture of odors gets mapped into the sum vector of all the components, normalized again to unit length. Finally, the angle of this vector gets corrupted by some perceptual noise. Two vectors around the circle will be discriminable if they are separated by more than the noise.

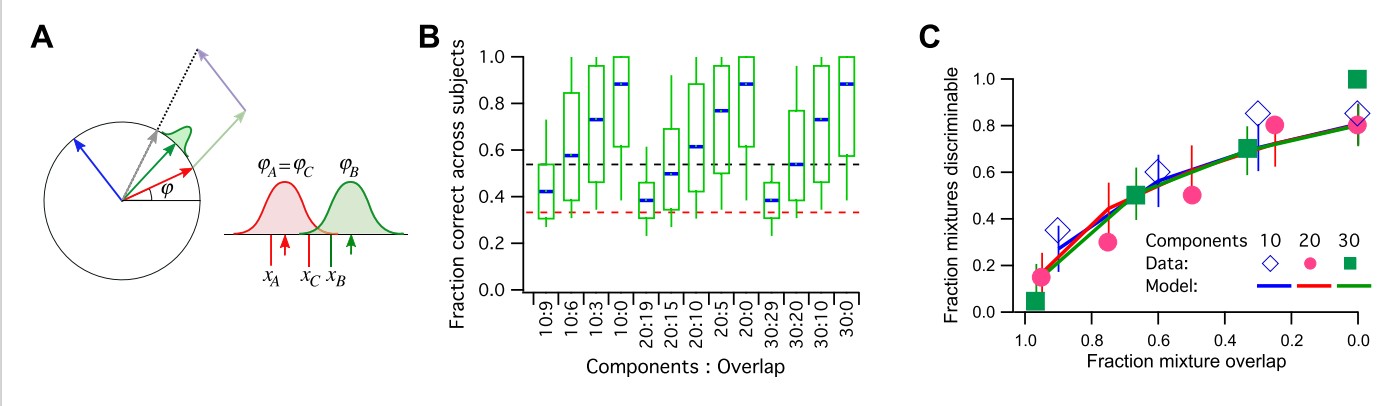

**Figure 3**. A model simulation of the human smell experiments. (**A**) Left: Each primary odor gets mapped into a unit vector (e.g., red, green, blue). Mixtures of odors get mapped into the normalized sum vector (gray). Right: When a subject sniffs an odor vial, the odor angle is corrupted by Gaussian noise and a value is drawn from that distribution. Here three vials were presented, two (**A** and **C**) containing the identical odor and a third (**B**) a different odor. This produced response variables $x_A$, $x_B$, and $x_C$. On this trial, $x_B$ and $x_C$ are closest to each other, so the subject (incorrectly) identifies **A** as the odd odor. (**B**) Discriminability of odor mixtures under this model (compare to *Bushdid et al., 2014*, Figure 2C). Mixtures were simulated according to the reported procedure with 10, 20, or 30 components and varying overlap. Each mixture pair was presented to 26 subjects, and the fraction of correct identification determined across subjects. Box-and-whisker plot shows the distribution of that fraction with percentiles 10, 25, 50, 75, 90. Average over 1000 repeats of the procedure with different random numbers. Red dashes: chance performance. Black dashes: criterion for discriminability (14/26 correct). (**C**) Fraction of discriminable mixtures as a function of their overlap (compare to *Bushdid et al., 2014*, Figure 3D). This is the fraction of mixture pairs in each class that exceeds 50% correct identification across subjects (above the black line in panel **B**). Lines are mean ± SD. Symbols are data from *Bushdid et al. (2014)*. The model used Gaussian noise with a SD of 0.4 radians.

This encoding model can simulate the reported human subject experiments in detail (*Figure 3B,C*): For every 'odd-man-out' trial, three mixtures are presented to the model, of which two are identical and the third is a different target mixture. The model maps all three onto the unit circle, adds noise to each, and asks which two are closest to each other. Then it reports the more distant one as the odd odor. If that corresponds to the target odor, it is a 'correct' decision (*Figure 3A*). Qualitatively, mixtures that share a large fraction of components produce sum vectors at nearby angles, which makes them less discriminable because of the perceptual noise (*Figure 3B*, compare to Figure 2C of *Bushdid et al., 2014*). Again one can compute the fraction of discriminable odor mixture pairs, namely those correctly identified in >50% of all trials.

In *Figure 3C*, I plot the results of this simulation along with the published data (from Figure 3B of *Bushdid et al., 2014*). The model has a single parameter, namely the amount of perceptual noise; nothing else is adjustable. With a noise value of 0.4 (SD of the Gaussian noise measured in radians) the fit to the data is quite good. Almost all the data points are within the 1 SD error bars. At noise = 0.3 the model does somewhat better than the humans, at noise = 0.5 somewhat worse.

For a noise value of 0.4, how many mutually discriminable stimuli are there? We should use the standard of *Bushdid et al. (2014)* for discriminability of an odor pair: 50% correct identification in the 'odd-man-out' odor test. With that criterion, one can place at most 10 vectors around the unit circle such that each is discriminable from its neighbors. Therefore, the published measurements are consistent with a model in which humans can distinguish just 10 olfactory stimuli from each other.

Many of us have experienced more than 10 different smells, so these experiments did not come close to exploring the richness of human olfactory experience. Why is the lower bound from this odor psychophysics study so weak? Actually, the result is just about expected from the effort expended. To confirm that 10 odors are mutually discriminable by brute force one has to compare each odor to every other one. The authors did 260 pairwise comparisons, of which only about half were discriminable. So one expects to find evidence for about $\sqrt{148}$ distinct odors, close to what was obtained.

The ring model of *Figure 3* assumes that odor percepts lie in a 1-dimensional space. If one allows for higher numbers of dimensions, then the predicted number of discriminable odors increases, approximately exponentially with the dimensionality of the perceptual space (L Abbott, E Schaffer, and R Axel, personal communication). Since we do not know the dimensionality of odor space, the

results of *Bushdid et al. (2014)* are equally consistent with 10 or a trillion discriminable odors or anything in between. In other words, the experiments fail to distinguish alternative hypotheses for the number of discriminable odors, even absurd ones that are clearly incorrect.

## Where is the flaw?

In the following discussion I will use the term 'percept' to strictly mean *the internal state of the sensory system at the stage where discrimination decisions are made*. For us humans two things have the same percept when they look or smell the same. I will extend the term to toy models, bacteria and mice, without presuming that those creatures engage in the more contemplative aspects of perception. Two stimuli may be physically distinct, yet cause the same percept, and thus become indiscriminable, as is the case for many color spectra. Two stimuli can even be distinct at the level of sensory receptors, and still produce the same percept, as in cases to be discussed below.

The failure of the method in *Bushdid et al. (2014)* occurs after the sphere-packing estimate (*Figure 1B*). It involves a step that is never mentioned but implicit in the procedure: the assignment of odor percepts to the different spheres in stimulus space. There are at least two problems that can lead to enormous overestimates of the number of discriminable odors, First, the authors assume that every one of the spheres packed into the space corresponds to a different odor percept. But this is unwarranted. The measurement of critical distance for discrimination only ensures that *neighboring* spheres correspond to different percepts, it says nothing about more distant spheres. Thus the same odor percept may recur over and over again for physically distinct odor mixtures.

The situation can be understood already in 2 dimensions. *Figure 4A* shows a set of close-packed pennies on the desktop. Every penny is different in color from its neighbors. More importantly every penny in the whole space has a different color from every other penny. If the pennies correspond to odor percepts one could legitimately say that this organism discriminates as many odors as there are pennies. However, there are many other ways to color close-packed pennies so that no adjacent ones have the same color. In fact, three colors are sufficient (*Figure 4B*), so just three percepts could account for an infinite number of spheres in a 2-dimensional odor space.

One might object that this is an esoteric arrangement: What kind of olfactory system would produce this strange periodic recurrence of the same percept? However, there is a perfectly natural color progression that also provides enormous savings of colors. In the arrangement of *Figure 4C* the percept stays the same along one dimension of the space and varies along the orthogonal dimension, yet all neighboring spheres are distinct. The number of pennies one can pack this way goes as the square of the number of colors available. In 128-dimensional space, the number of hyperspheres one can pack this way will go as the 128th power of the number of percepts.

So the key logical flaw is the assumption that all the close-packed hyperspheres produce different odor percepts. One can trace this back to the unstated assumption that the odors should systematically become more discriminable at greater distances in this space, *and equally in every direction*. Here I showed that there is a simple and natural way to violate that assumption, namely if the percept depends only on a single function of the coordinates. Then the odors become more discriminable with distance along one dimension, but remain indiscriminable along all other dimensions. More generally, if the percepts live in some low-dimensional space (as is the case, e.g., for color vision and for the hearing of pure tones) and then one embeds that space in 128 dimensions, this will produce a similarly efficient labeling of the close-packed spheres. This is what causes the astronomical overestimates in the analysis of *Bushdid et al. (2014)* when applied to the three model simulations above.

The above arguments cannot fully explain the results obtained with the model of the 3-state microbe. Here there are only three possible percepts ('yum', 'yuck', and 'meh') that can be used to paint the hyperspheres. Yet, in 128-dimensional space every sphere has at least 256 nearest neighbors. Clearly it is impossible to distinguish a sphere from all its nearest neighbors with just three colors. This brings us to a second

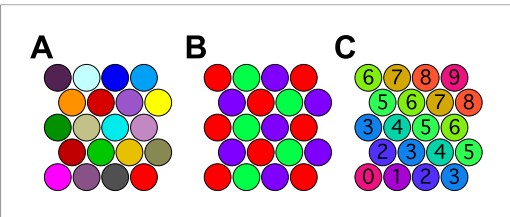

**Figure 4**. Coloring close-packed spheres in 2 dimensions so no nearest neighbors have the same color. (**A**) All colors different. (**B**) Three colors suffice. (**C**) Progression of colors in one dimension only.

weakness of the method: the definition of the critical distance *D*. This is taken as the distance at which 50% of odor mixture pairs are discriminable. In other words, two points separated by *D* in odor space only need to produce a different percept 50% of the time. So when assigning percepts to the spheres in 128-dimensional space, we merely have to ensure that each sphere is distinct from 50% of its neighbors. That can be done trivially with just two colors, painting alternating spheres white and black. That makes every sphere different from 50% of all its neighbors. The 3-percept microbe can do a bit better, which is why discriminability rises to almost 2/3 at large distances (*Figure 1A*).

Would it help to raise the criterion for discriminability to a higher percentage of mixture pairs? If one raises it to 90% then 10 odor percepts are sufficient to color the entire space of spheres. A popular estimate for the number of discriminable odors (though without solid scientific basis, as reviewed in *Bushdid et al., 2014*) is around 10,000. To ensure that a sphere-coloring system uses at least 10,000 odors, one would need to raise the criterion for discriminability in the definition of *D* to 99.99%. So there needs to be a data point in Figure 3B of *Bushdid et al. (2014)* with an ordinate of 99.99%. This would require human subjects to perform several tens of thousands of pairwise comparisons in just one mixture class, a truly extraordinary experimental effort.

## Discussion

### How to proceed?

The arguments above illustrate that the mathematical method for exploring sensory spaces advocated in *Bushdid et al. (2014)* does not work. Fundamentally the extrapolation presumes that the space of odor percepts has at least 128 dimensions. Furthermore, the 128 primary odors chosen must represent 'orthogonal' directions in that space, so that the percept varies with distance in the same manner and independently along each of these dimensions. Nothing in the report suggests why we should believe this, and the assumptions seem implausible *a priori*. Violation of those assumptions, for example if the true dimensionality of odor space is much less than 128, leads to dramatic failure of the estimate. How can one do better?

To obtain the number of discriminable odors, we need to determine the largest set of stimuli such that every stimulus can be discriminated *from every other one*, not just from nearest neighbors. Classic studies of color vision and tone hearing have achieved that. For color vision, the answer is approximately $n = 1$ million. How can one come to that conclusion without doing $n^2$ discrimination tests? This is possible only by taking advantage of the systematic structure of the perceptual space. For example, if light *A* looks more red than light *B* and *B* looks more red than *C*, then one can trust that *A* will look more red than *C*. The same applies in tone hearing: If tone *A* sounds lower than tone *B*, and *B* sounds lower than *C*, then *A* will sound lower than *C*. This transitivity of discrimination means that it is sufficient to measure discrimination of neighboring stimuli to ensure mutual discrimination of all stimuli in the set, which reduces the experimental burden from order($n^2$) down to order($n$). Effectively these studies work in a low-dimensional subspace of all stimuli (R, G, B for lights; frequency and amplitude for pure tones), within which the progression of the percepts is monotonic, such that transitivity applies. So success of these percept-counting studies relied entirely on recognizing early on that the perceptual space is low-dimensional.

### The dimensionality of odor space

How to apply those insights to olfaction? Clearly one needs to first answer 'how many dimensions span the perceptual space of odors'? The number of 400 human odorant receptor types certainly sets an upper bound on the dimensionality of the perceptual space. And one commonly hears the argument: 'Why would Nature make so many receptors, unless it is to discriminate all possible patterns of activation'? However, there may be another explanation: We need this many receptors simply to *sense* the molecules of interest, not to disambiguate all possible mixtures of those molecules.

The bacterium *Escherichia coli* illustrates this idea: It has five different chemoreceptor proteins with different ligand spectra. In principle, *E. coli* could therefore analyze odor mixtures in 5-dimensional space and react differently to every possible mixture. But it doesn't. The outputs of all five receptors readily converge on a single variable, namely the concentration of the CheY signaling molecule (*Grebe and Stock, 1998*). This single variable controls the bacterium's locomotor decisions, so we may identify it with the odor percept in the present use of the term. *E. coli* projects the 5-dimensional

receptor space onto a 1-dimensional perceptual space of attraction/repulsion. Why then does it need five receptors, including different receptor proteins for aspartate and for serine, both amino acids? Presumably it is difficult to make a generic amino acid receptor with high sensitivity. Both the amino and the carboxyl ends of the molecule vary in their charge distribution depending on pH, and a single binding pocket directed at these regions would not offer sufficient affinity under all conditions. Instead, the two receptor proteins focus on other more stable portions of the ligand, but those are also unique between serine and aspartate.

The same arguments apply to odorants in the human nose. The molecules of interest are there at micromolar concentrations or less, in a mucus soup of other components at millimolar concentration. To sense the odorants separately from the mucus, a receptor needs to bind them with high affinity. That means many contact sites between binding pocket and ligand, which in turn leads to selectivity for the shape of the ligand. Even if the olfactory system just wanted to distinguish odors along one dimension (attractive/repulsive), it would be impossible to make a receptor that is selective just for the attractants. Instead Nature makes many receptors that are each selective for small groups of related molecules and then combines their signals appropriately using the nervous system.

In this picture the dimensionality of receptor space is determined by molecular principles involving the number of ligands of interest, their relevant concentrations, the energetics of ligand binding, and the design limitations of protein structures. The dimensionality of perceptual space, on the other hand, is governed by behavioral and ecological constraints: the nature of olfactory cues in the environment, the kinds of decisions the animal makes based on odorants, and the need to associate new odors with unusual events. There is no principled reason that this perceptual space should have the same dimensionality as the receptor space. And we have the neural circuits of the olfactory system to create an arbitrary map from one space onto the other.

Another sensory system serves to illustrate this difference between receptor space and perceptual space: touch. Every hair follicle on our skin contains a sensitive mechanoreceptor, several million altogether (*Zimmerman et al., 2014*). This makes us highly sensitive to touch: we can reliably detect the bending of an individual hair almost anywhere on the body. But clearly we cannot discriminate all patterns of bent hairs. Brushing across your head twice in a row feels very much the same, even though it is certain to cause two different patterns of activity among the touch receptors. As for *E. coli* chemotaxis, there is a benefit to detecting many possible sensory inputs, but no need to discriminate all possible patterns of those inputs.

Beyond arguments by analogy, some recent studies of human olfaction suggest that the perceptual space for odors may have rather few dimensions. Furthermore, the dominant axes of this perceptual space can be related systematically to the physical characteristics of odorous molecules (*Secundo et al., 2014*). Another relevant observation is that mixtures containing many (>20) diverse odorants tend to smell alike, even if they don't share any molecular components (*Weiss et al., 2012*), a phenomenon that has been termed 'olfactory white', in analogy to the 'white' percept associated with a mix of many colored lights. This suggests that the dimensionality of odor percepts may be around 20 or less. *Table 1* summarizes the number of dimensions of various spaces discussed here.

A useful experimental approach might then be to rigorously measure the dimensionality of perceptual space *at least at one point*. For example, choose $N$ primary odors, and consider arbitrary mixtures of those as the odor space. Define the 'white point' as the mixture of all those odors at half concentration, that is, the odor vector w = (0.5,…,0.5). How does odor perception vary as the stimulus deviates a little from this point? To first order, the discriminability $d$ between the white odor at w and an odor at w + x will vary as a quadratic form of the deviation vector x, namely: $d = x^T S x$. We want to know the sensitivity matrix S. By definition it is positive definite and symmetric, and thus has $N(N + 1)/2$

**Table 1**. Number of dimensions of various spaces involved in sensory discrimination

|  | Toy microbe | Ring model | Human color | *E. coli* smell | Human smell |
|---|---|---|---|---|---|
| Stimuli | ∞ | ∞ | ∞ | ∞ | ∞ |
| Receptors | ∞ | ∞ | 3 | 5 | ~400 |
| Percepts | 1 | 1 | 3 | 1 | 1–20? |

The symbol ∞ stands for 'very large or infinite'.

unknown components. So one needs to measure the just-discriminable-distance along $N(N + 1)/2$ directions from the white point. Clearly this is an experimental challenge, but it seems plausible at least for $N = 20$. If so, then the structure of the matrix S can reveal the dimensionality: In particular, if it has just a few large eigenvalues, those identify the relevant directions in odor space. By contrast, if all eigenvalues are comparable, then the perceptual space has dimensions higher than $N$.

Animal studies can play an important role here. Mice are readily trained to distinguish odors, even closely related mixtures. More importantly, they offer an opportunity to stimulate the receptor neurons directly, by optogenetic activation of the olfactory bulb (*Spors et al., 2012*). In a suitably engineered animal one could drive arbitrary activation patterns of the different olfactory receptor types by shining patterned light onto the glomeruli in the olfactory bulb. This approach promises several benefits in a study of odor dimensions: First, it does away with the tedium of olfactory stimulation, such as mixing dozens of vapors, switching valves, flushing tubes, and waiting for odors to dissipate. Using light, a different combination of receptors can be driven with millisecond precision and at high repetition rates. Also, this method allows patterns of stimulation that may not ever occur with natural odorants; one could then test if the perceptual space differs from receptor space, and is shaped to the ecology of real odors.

In a way, one can think of the olfactory bulb surface as a retina for the smell system. Olfactory objects produce spatio-temporal patterns on this surface, and the downstream neural circuits are busy identifying, discriminating, or learning those spatio-temporal patterns. The optogenetic approach simply takes the analogy one step further by using light as a stimulus. At the same time, there is a parallel effort ongoing in vision science to determine the dimensionality of human pattern vision. It is clear already that the number of dimensions is much lower than the number of receptors on the retina. One can make pairs of visual images that have very different effects on the retina, but look the same to human subjects (*Freeman and Simoncelli, 2011*). And a systematic approach to measuring dimensionality of pattern vision is beginning to yield results (JD Victor, personal communication). Based on these developments, I suggest that pursuing the analogy of smell to pattern vision will be more fruitful than the analogy to color vision.

Regardless of approach though, determining the dimensionality of the space of odor percepts is a precondition to estimating the number of distinct percepts. The recognition that color space is three-dimensional has had enormous impact in science, art, and technology, as anyone reading this on a color monitor will confirm. The search for a similar basis set for odors has fascinated scientists, engineers, and perfumers for some time (*Gilbert, 2008*). Even proving that a low-dimensional basis does not exist would be a major advance.

## Materials and methods

All simulations and graphics were produced with Igor (Wavemetrics, Tigard, OR). Annotated code is available as a supplement to this article (*Source Code 1*). Simulation of the three models (*Figures 1–3*) followed the same process as the human odor tests performed in *Bushdid et al. (2014)*:

1. Selection of the 128 primary stimuli. These were drawn at random from the stimulus space. *Figure 1*: binary distribution over (−1, +1). *Figure 2*: Uniform distribution of R, G, and B over [0, 1], followed by normalization to a length of 1/30. *Figure 3*: Uniform distribution on the unit circle. These are conservative choices, in that a random set of primaries does not cover the stimulus space particularly well, and thus will produce mixtures that occupy only a portion of the space. This will therefore underestimate the number of discriminable stimuli. By contrast Bushdid et al. chose odors that were 'well distributed in both perceptual and physicochemical stimulus space'.
2. Creation of mixtures. Pairs of mixtures containing $N$ primaries of which $O$ are shared were created by choosing at random $2N-O$ components from the set of primaries, summing the first $N$ to mixture 1, and the last $N$ to mixture 2. For every class of mixtures (i.e., combination of $N$ and $O$), 20 mixture pairs were created. The rules for combining the stimulus values in a mixture were as follows: *Figure 1*: Simple addition of the binary stimulus values. *Figure 2*: Addition of the primary vectors. *Figure 3*: Addition of the unit vectors followed by normalization to unit length. This normalization emulates the elimination of odor intensity cues in *Bushdid et al. (2014)*.
3. Discrimination test. Every pair of mixtures was presented to the model to determine whether it was discriminable or not. *Figure 1*: The model classifies every mixture of $N = 30$ odors into the percepts 'yum', 'meh', or 'yuck' depending on whether the sum of stimulus values is >2, from 2 to −2, or <−2 respectively. Two mixtures that fall into different classes are discriminable. *Figure 2*: The model performs an odd-man-out discrimination test among three samples as done for human subjects in

*Bushdid et al. (2014)*. Two of the samples contain mixture 1 and the third contains mixture 2. Perceptual noise was simulated by adding a Gaussian random variable to each of the three coordinates of the mixture vectors. The resulting three sample vectors are inspected and the two with the smallest distance are declared to be the same. For every mixture, this is performed 26 times, with different draws from the noise distribution. A mixture pair is declared discriminable if the model gives the correct response on >50% (14 or more) of those trials (note chance performance is 1/3). In *Bushdid et al. (2014)* these 26 trials were performed by 26 different human subjects. There was some indication that different subjects had different abilities, but the analysis merged them all. In my simulation, all trials were done with the same amount of perceptual noise. The noise magnitude was chosen so that stimuli separated by a distance of 0.01 are just discriminable. This leads to ~1 million distinct percepts in the RGB stimulus space, a conservative choice, because the empirical estimates of that number are somewhat larger. *Figure 3*: This followed the same odd-man-out procedure as for *Figure 2*. Perceptual noise was simulated by adding a Gaussian random variable to the angle of the mixture vector (*Figure 3A*).

4. Discriminability of mixture classes. For each class of mixtures I computed what fraction of the 20 were discriminable (by the criteria in 3), and plotted this against the mixture overlap (*Figures 1A*, *2B,C*, *3C*). To estimate the reliability of the simulation, the entire procedure was repeated 1000 times (with different random numbers) and the plots show the mean and standard deviation of the outcome. *Figure 3C* shows my simulation along with the data from the Bushdid et al.'s human experiments.

5. Estimating the number of discriminable stimuli. From the graph of discriminability vs overlap, the critical distance $D$ was taken to be the number of unshared stimuli that allows 50% discriminability in that class of mixtures. The number of regions of diameter $D$ that can fit into the stimulus space was computed using the formula presented in *Bushdid et al. (2014)* (*Figure 1C*). If there are $C$ primaries total and $N$ primaries per mixture, then the number $S$ of such regions is claimed to be

$$S = \frac{\binom{C}{N}}{\sum_{R=0}^{D/2}\binom{N}{R}\binom{C-N}{R}}.$$

## Acknowledgements

Many thanks to Adam Shai for extended discussions.

## Additional information

### Funding

No external funding was received for this work.

### Author contributions

MM, Conception and design, Analysis and interpretation of data, Drafting or revising the article

## Additional files

### Supplementary file

• Source code 1. Annotated Igor (Wavemetrics) code.

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
