## [Decision Letter]

Thank you for submitting your work entitled “On the dimensionality of odor space” for peer review at *eLife*. Your submission has been favorably evaluated by Eve Marder (Senior editor) and three reviewers, one of whom is on our Board of Reviewing Editors.

The following individuals responsible for the peer review of your submission have agreed to reveal their identity: Axel Borst (Reviewing editor); Frédéric Theunissen (peer reviewer). A further reviewer remains anonymous.

The reviewers have discussed the reviews with one another and the Reviewing editor has drafted this decision to help you prepare a revised submission.

Summary:

In this manuscript, Meister re-analyzes a paper by [1], that claims that humans can discriminate one trillion odors. He shows by different methods that the claim is substantially flawed. To make this point, Meister uses different approaches. In the most immediately convincing approach, he constructs a model organism with 3 different percepts and shows that, subjected to the identical test performed by Bushdid et al., one concludes that the organism can discriminate one trillion odors: however, we know by construction that it only has three odor percepts. I repeated the model simulations myself just to confirm these results. Meister further analyzes the method and demonstrates convincingly that the mistake ‘in depth’ made by Bushdid et al. is that it equates the number of discriminable odor mixtures with the number of volumes that can be packed in high-dimensional odor space as calculated from the average minimal distance in odor space: it is never shown that each of these mixtures represents a unique percept in perceptual space. In the final chapter, he discusses how to measure the dimensionality of perceptual odor space, making an argument that the large number of odorant receptors might exist in order to sense the large number of different odor molecules, rather than to discriminate each one of them from each other. He furthermore advocates for using an optogenetic approach in rodents to avoid a lot of problems associated with producing and handling odorant mixtures.

The paper is well written and the toy examples intuitively demonstrate the flaw. All reviewers really enjoyed reading the manuscript and found it to be a very nice contribution worth being published in *eLife*. However, there are some remarks made by the reviewers that should be addressed before the paper can be published.

Essential revisions:

1) The issue requires great clarity and consistency of definitions. In particular, it concerns a number of different spaces: the space of chemical stimuli, the space(s) of neural activity, “perceptual” space and “action/behavioral” space. Meister sometimes interchanges terminology in such a way that a straight line of thought becomes hard to trace. For example, are “percepts” and “discriminable stimuli” the same thing? In the bacterial example, Meister calls the reactions alternatively “responses” and “percepts”. This quibbling about words does not change the bottom-line of Meister's argumentation, but non-defined terminology weakens it by exposing it to potential counter-arguments. The reviewers suggest to clearly define all the important words used throughout.

2) The next comment concerns Meister introducing perceptual space for color vision as being 3 dimensional, and those dimensions as being defined by the response spectra of the 3 types of cones found in trichromatic primates. But is this not a shortcut between stimulus-, neural- and perceptual spaces? The dimensions of color perception are, by some classical definitions at least, hue, lightness and saturation. In this space, colors live in a sort of spindle-shaped subspace in which perceptual hue, for example, seems to loop back onto itself, making the mapping between stimulus (say wavelength spectrum) and percept non-trivial (technically speaking, non-homeomorphic). The 3-dimensional space that Meister considers is a neural space, defined by photoreceptor subtypes, and not directly the perceptual space of color or hue. Both happen to be low dimensional, which is probably not a coincidence, and this does not change anything to the importance of quantifying humans' perceptual abilities for color discrimination; but the distinction is important because of the non-trivial nature of the mapping between these different spaces. In fact the topology-of-perceptual-space argument is related to that used by Meister in his Figure 4 and corresponding text. We suggest rewording, so as not to weaken the otherwise valid criticisms. Also, color vision in many animals starts with many more cone types (up to 12 in some southern seas crustaceans). No one really knows how fine or structured color discrimination is in those brains. In the same vein, some human cultures name colors in ways that differ dramatically from ours. The general issue of color perception is thus potentially more complex still, and nothing indicates that the geometry of odor-perception space has much to do with that of color-perception space. Finally there is a great deal of O.R. gene polymorphism among humans, presumably increasing local variance in particular regions of odor-perception space. All this to say that, even though Bushdid et al. made regrettable methodological mistakes, the existence of a relatively simple and well-bounded solution to the problem of trichromatic vision does not guarantee the existence of similar solutions to the problem of color vision in general, or to olfaction. The critique should therefore focus better on the mathematical/statistical mistakes of Bushdid et al.

3) The third point concerns the bacterial example (“Failure on a simple 3-percept model”). Here, we have a split opinion. While two reviewers actually like the example as it is, one reviewer says the following: “I understand the aim of the example; I understand the need to illustrate the fundamental difficulty of bounding an estimate with small samples; but this particular example ends up looking like a specious construct, based on a trick. The trick is to map all possible discriminable items into three classes (so called: yuck, meh and yum), which Meister calls alternatively ‘responses’ and ‘percepts’. But ignoring for an instant that bacteria probably lack perceptual abilities, the only reason why the discriminating ability of the bacterium is so low is because of a definition that need does not apply to a real nervous system. While I understand the goal of this chapter, I think it is too facetious to be convincing.” We leave it up to the author to decide.

---

## [Author Response]

*1) The issue requires great clarity and consistency of definitions. In particular, it concerns a number of different spaces: the space of chemical stimuli, the space(s) of neural activity, “perceptual” space and “action/behavioral” space. Meister sometimes interchanges terminology in such a way that a straight line of thought becomes hard to trace. For example, are “percepts” and “discriminable stimuli” the same thing? In the bacterial example, Meister calls the reactions alternatively “responses” and “percepts”. This quibbling about words does not change the bottom-line of Meister's argumentation, but non-defined terminology weakens it by exposing it to potential counter-arguments. The reviewers suggest to clearly define all the important words used throughout*.

Yes, this vocabulary needs to be normalized. I made the following changes:

Left out reference to “perception” where it's not necessary, e.g. in the section on the toy microbe.

Precisely defined my use of the term “percept” where it starts to become useful (section “Where is the flaw?”).

Distinguished different kinds of spaces: stimuli, receptors, percepts. See text around the new Table 1.

*2) The next comment concerns Meister introducing perceptual space for color vision as being 3 dimensional, and those dimensions as being defined by the response spectra of the 3 types of cones found in trichromatic primates. But is this not a shortcut between stimulus-, neural- and perceptual spaces? The dimensions of color perception are, by some classical definitions at least, hue, lightness and saturation. In this space, colors live in a sort of spindle-shaped subspace in which perceptual hue, for example, seems to loop back onto itself, making the mapping between stimulus (say wavelength spectrum) and percept non-trivial (technically speaking, non-homeomorphic). The 3-dimensional space that Meister considers is a neural space, defined by photoreceptor subtypes, and not directly the perceptual space of color or hue. Both happen to be low dimensional, which is probably not a coincidence, and this does not change anything to the importance of quantifying humans' perceptual abilities for color discrimination; but the distinction is important because of the non-trivial nature of the mapping between these different spaces. In fact the topology-of-perceptual-space argument is related to that used by Meister in his*
Figure 4
*and corresponding text. We suggest rewording, so as not to weaken the otherwise valid criticisms. Also, color vision in many animals starts with many more cone types (up to 12 in some southern seas crustaceans). No one really knows how fine or structured color discrimination is in those brains. In the same vein, some human cultures name colors in ways that differ dramatically from ours. The general issue of color perception is thus potentially more complex still, and nothing indicates that the geometry of odor-perception space has much to do with that of color-perception space. Finally there is a great deal of O.R. gene polymorphism among humans, presumably increasing local variance in particular regions of odor-perception space. All this to say that, even though Bushdid et al. made regrettable methodological mistakes, the existence of a relatively simple and well-bounded solution to the problem of trichromatic vision does not guarantee the existence of similar solutions to the problem of color vision in general, or to olfaction. The critique should therefore focus better on the mathematical/statistical mistakes of Bushdid et al*.

The reasons to introduce human color vision are twofold: (1) That's what Bushdid et al. did. Their Abstract already claims that the human color vision pales compared to smell. I show that there's no basis for this claim. (2) This is a sensory system where we understand the physical/receptor/perceptual spaces well enough to answer the question posed by Bushdid et al.: how many discriminable stimuli are there? Tracing how this was accomplished offers a useful starting point for trying to do it in another sensory system. There is of course no expectation that the results of the method will be the same when applied to olfaction.

The descriptors “hue/saturation/lightness” are directly related to “R,G,B” (see e.g. Equation 3 in [6]). But the details of how humans describe colors (e.g. hue/saturation/lightness or various color words) are not relevant to the problem. The pairwise discrimination tests (two colors presented side by side in a split field) don't require that subjects have a color name or any other descriptor for what they see. Clearly we can discriminate many more colors than we can name. The same is true for odors.

*3) The third point concerns the bacterial example (“Failure on a simple 3-percept model”). Here, we have a split opinion. While two reviewers actually like the example as it is, one reviewer says the following: “I understand the aim of the example; I understand the need to illustrate the fundamental difficulty of bounding an estimate with small samples; but this particular example ends up looking like a specious construct, based on a trick. The trick is to map all possible discriminable items into three classes (so called: yuck, meh and yum), which Meister calls alternatively ‘responses’ and ‘percepts’. But ignoring for an instant that bacteria probably lack perceptual abilities, the only reason why the discriminating ability of the bacterium is so low is because of a definition that need does not apply to a real nervous system. While I understand the goal of this chapter, I think it is too facetious to be convincing.” We leave it up to the author to decide*.

I still think this toy model plays a useful role as a positive control. The number of discriminable odors is known a priori (3) and the calculations are very easy. More importantly, the model has all the necessary components: it treats mixtures of many stimuli, can compare pairs of such mixtures, and exhibits a clear performance limit in their discrimination. The performance limit is implemented by discretizing stimuli into a small number of classes, as opposed to using a continuous variable corrupted by noise. I fail to see anything facetious or tricky about this simplification. I added a few sentences of introduction and conclusion to this section.